# Enzymes with Lactonase Activity against Fungal Quorum Molecules as Effective Antifungals

**DOI:** 10.3390/biom14030383

**Published:** 2024-03-21

**Authors:** Elena Efremenko, Aysel Aslanli, Maksim Domnin, Nikolay Stepanov, Olga Senko

**Affiliations:** Faculty of Chemistry, Lomonosov Moscow State University, Lenin Hills 1/3, Moscow 119991, Russia

**Keywords:** lactamase, lactonase, organophosphate hydrolase, butyrolactone I, γ-heptalactone, ATP concentration, yeasts, filamentous fungi, quorum sensing, suppression

## Abstract

Since the growing number of fungi resistant to the fungicides used is becoming a serious threat to human health, animals, and crops, there is a need to find other effective approaches in the eco-friendly suppression of fungal growth. One of the main mechanisms of the development of resistance in fungi, as well as in bacteria, to antimicrobial agents is quorum sensing (QS), in which various lactone-containing compounds participate as signaling molecules. This work aimed to study the effectiveness of action of enzymes exhibiting lactonase activity against fungal signaling molecules. For this, the molecular docking method was used to estimate the interactions between these enzymes and different lactone-containing QS molecules of fungi. The catalytic characteristics of enzymes such as lactonase AiiA, metallo-β-lactamase NDM-1, and organophosphate hydrolase His_6_-OPH, selected for wet experiments based on the results of computational modeling, were investigated. QS lactone-containing molecules (butyrolactone I and γ-heptalactone) were involved in the experiments as substrates. Further, the antifungal activity of the enzymes was evaluated against various fungal and yeast cells using bioluminescent ATP-metry. The efficient hydrolysis of γ-heptalactone by all three enzymes and butyrolactone I by His_6_-OPH was demonstrated for the first time. The high antifungal efficacy of action of AiiA and NDM-1 against most of the tested fungal cells was revealed.

## 1. Introduction

Due to the accumulation of information about microscopic fungi capable of showing resistance to the effects of fungicides used in practice today, a serious problem of fungal lesions that are dangerous to human health, animals, and crops is being formed [1,2,3]. In October 2022, the World Health Organization published a report in which, for the first time, a list of fungal “priority pathogens” was presented, which includes 19 fungal cultures that pose the greatest threat to public health [3].

Today, five main classes of fungicides (polyenes, pyrimidine analogues, azoles, allylamines, and echinocandins) are being discussed for practical use. These fungicides do not have a specific action and are used in fairly high concentrations to suppress the growth of cells of microscopic fungi, thereby causing the development of resistance in fungi. It is evident that there is a need to find and use other approaches and solutions that would lead to the death of fungal cells or at least effectively, and for a long time, inhibit their growth and metabolic activity [4,5].

It is known that widely used chemically synthesized fungicides have higher ecotoxicity than antifungal preparations of natural origin. In this regard, the main interest in the search for new antifungal solutions is shifting towards various natural, biologically active compounds. At this stage, plants are considered primarily as sources for the isolation of such substances [6,7]; however, their action is also not selective and, if used, can affect (inhibit) different metabolic and biochemical processes not only in fungal cells.

One of the main mechanisms of resistance development in fungi is analogous to that found in bacteria [8], and is the so-called quorum sensing (QS). QS is the ability of cells to communicate with each other in a population due to the synthesis and secretion of small signaling molecules into the extracellular environment [9]. It depends on the concentration of the cell population and can stimulate the expression of so-called silent genes, the synthesis and secretion of virulence factors, etc. Among the QS signaling molecules in fungi, various lactone-containing compounds (γ-butyrolactone, γ-heptalactone, butyrolactone I, etc.) were identified [10,11] (Figure 1 and Appendix A).

In this regard, one of the possible approaches to overcoming fungal resistance is the destruction of these lactone-containing signaling molecules of fungi using enzymes exhibiting lactonase activity similar to that used in the case of quorum quenching in bacteria [4,12].

By now, the efficient use of lactonases for the destruction of lactone-containing signaling molecules of bacterial QS, N-acyl homoserine lactones (AHLs), is actively studied [4,13]. In contrast, to the best of our knowledge, to date, only a limited number of studies are known to be aimed at the use of lactonases against QS signaling molecules of fungi [14,15]. When conducting research in this direction, we previously found that the enzyme hexahistidine-containing organophosphate hydrolase (His_6_-OPH) copes well with a similar task [12,16], catalyzing the hydrolysis of AHL molecules of various structures.

It is also known from the literature that various bacterial lactonases (AiiA, AiiB, AidC, AaL) are able to effectively hydrolyze different AHL molecules [14,17,18,19,20]. It has recently been shown that representatives of the superfamily of metal-containing β-lactamases (MBLs) have an active-site structure that is similar to lactonases and are able to hydrolyze AHL molecules [21,22] (Table 1 [14,15,16,17,18,19,20,21,22,23,24]). The reactions catalyzed by these enzymes exhibiting lactonase activity consist of the opening of the lactone ring and are similar to the β-lactam ring rupture reaction, which is typical for catalysis based on the action of MBLs [25,26].

Thus, the purpose of this work was to investigate the effectiveness of the action of enzymes exhibiting lactonase activity against lactone-containing molecules of various fungal cells. It was decided to initially evaluate the possibility of using of enzymes capable of hydrolyzing lactone-containing bacterial quorum signaling molecules for the destruction of analogous fungal quorum molecules in silico.

For this, the molecular docking method was used to investigate the interactions between the enzymes exhibiting lactonase activity listed in Table 1 and the quorum molecules of different fungi (Figure 1). The obtained data were used to determine the theoretical values, as well as the experimental catalytic characteristics of the enzymes selected based on the results of computer modeling, as objects for research. The antifungal activity of the selected enzymes was further evaluated against various fungal and yeast cells. The effectiveness of this antimicrobial effect was assessed by the residual intracellular concentration of adenosine triphosphate (ATP) using highly sensitive and specific bioluminescent ATP-metry [27], which has proven itself well in studies of various antimicrobials [28,29].

## 2. Materials and Methods

### 2.1. Materials

The lactone-containing quorum molecules γ-heptalactone and butyrolactone I were purchased from Sigma-Aldrich (Darmstadt, Germany). N-acyl homoserine lactonase (AiiA) was obtained from ProteoGenix (Schiltigheim, France). Recombinant bacterial metallo-β-lactamase NDM-1 was purchased from RayBiotech, Inc. (Norcross, GA, USA). His_6_-OPH enzyme was produced by a patented method using recombinant Escherichia coli strain SG13009[pREP4] (Qiagen, Hilden, Germany) transformed by a plasmid encoding His_6_-OPH, and, further, the enzyme was purified by a published procedure [30].

### 2.2. Computational Methods

Three-dimensional structures of enzymes were obtained from the Protein Data Bank. The known crystallographic structure of OPH (PDB ID: 1QW7) was used to prepare the structure of His_6_-OPH, which was further modified by His_6_-tag as described previously [12].

Ligand (lactone) structures were created using the ChemBioDraw software (ver. 12.0, CambridgeSoft, Waltham, MA, USA), and then ChemBio3D was used to apply energy minimization with force field MM2, and structures in the PDB format were obtained. Then, AutoDockTools (as part of MGLTools ver. 1.5.6, available at http://mgltools.scripps.edu/ accessed on 11 June 2023) was used to obtain structures in PDBQT format from PBD structures with atomic charges calculated with the Gasteiger–Marsili method [31].

Using AutoDock Vina (ver. 1.1.2, available at http://vina.scripps.edu/ accessed on 12 June 2023) and a desktop computer equipped with an Intel Pentium Dual-Core CPU E5400 with 2.7 GHz and 3 GB of available memory, enzyme–ligand complexes were calculated [32]. The grid box was approximately centered on the center of mass of the enzyme and its size was chosen so that any enzyme surface was within the box with an additional margin. Following the procedure, the “receptor” (i.e., enzyme) was proposed as rigid and the “ligand” (i.e., QS molecule) was fully flexible. Calculations were performed with default program options. The best 8 poses with minimal energy were selected in accordance with our previous studies with His_6_-OPH and the results obtained in the development of new effective antibacterials [28,29].

The solvent accessible area occupied by ligands on the surface of the enzyme was calculated using the “get_area” function of PyMOL (version 1.7.6.0).

### 2.3. Hydrolysis of Lactones

A lactone hydrolysis reaction was carried out as follows: 5–1000 μL of γ-heptalactone or 1–100 μL of butyrolactone I was dissolved in PBS buffer (pH 8.0). Reactions were initiated by the addition of AiiA (12 µg/mL), NDM-1 (2 µg/mL), or His_6_-OPH (100 µg/mL) enzymes or a buffer solution for controls and incubated at 37 °C or room temperature. Over time, aliquots of the reaction medium containing γ-heptalactone were sampled and used for GC/MS analysis.

GC/MS analysis was carried out using an Agilent 7820A chromatograph equipped with mass detector 5977B (Agilent Technologies, Santa Clara, CA, USA). Samples of the reaction medium were immediately extracted using 500 µL of ethyl acetate by vortexing for 1 min followed by centrifugation (13,000× *g*, 5 min). Then, 1 μL of ethyl acetate phase was injected into a DB-5 30 m capillary column (0.25 mm, 0.25 u, Agilent Technologies). The flow rate of the carrier gas (ultrahigh-purity-grade He) was 1.2 mL/min. The thermostat program was 70 °C (5 min) and heating at 15 °C/min to 220 °C (10 min). The range of the ion mass scan was 10–150 *m*/*z*. Electron ionization was at 70 eV. Chromatograms were analyzed using the ChemStation DA/MassHunter software (Agilent Technologies, version B.07.06.2704) with the NIST 2014 structural library.

Samples containing butyrolactone I were analyzed using the Agilent 8453 UV–visible spectroscopy system (Agilent Technology, Waldbronn, Germany) equipped with a thermostated analytical cell in the wavelength range from 150 nm to 600 nm.

The values of the Michaelis constant (*K*_m_) and the maximum rate of the enzymatic reaction (*V*_max_) were calculated by hyperbolic approximation using the least squares method in Origin Pro (ver. 8.1 SR3, OriginLab, Northampton, MA, USA). The obtained *K*_m_ and *V*_max_ values were further used to calculate catalytic constant (*V*_max_/E_0_) and action efficiency constant (*k_eff_* = *V*_max_/(E_0_ × *K*_m_) values for enzymatic activity.

### 2.4. Determination of Antimicrobial Activity

The antimicrobial efficiency of the action of enzymes was determined as described previously [29] with minor modifications. For this, the fungi *Rhizopus oryzae* F814, *Aspergillus niger* F679, *Trichoderma atroviride* F207, and *Fusarium solani* F819, as well as yeasts *Saccharomyces cerevisiae* Y-1234 and *Candida tropicalis* Y-2245, were used.

To accumulate biomass, the cells of fungi and yeasts were grown in the corresponding culture medium as described previously [29]. The fungal and yeast cells were cultivated using a thermostatically controlled Adolf Kuhner AG shaker (Basel, Switzerland) at 28 °C with constant stirring at 150 rpm.

All microorganisms were separated from the nutrition media after cultivation by centrifugation for 5 min at 10,000× *g* (Beckman Avanti J-25, Beckman Instruments Inc., Fullerton, CA, USA).

The growth of yeast cells was monitored with an Agilent UV-8453 spectrophotometer (Agilent Technology, Waldbronn, Germany) at 540 nm.

To determine the dry weight of the fungal cells gravimetrically, a weighed sample of the fungal cell biomass separated from the culture broth via centrifugation was brought to a constant weight by drying.

Further, the cell biomass was suspended in 9 g/L of NaCl prepared on the basis of a 50 mM phosphate buffer (pH 7.5). The concentration of the yeast cells was ca. (1 ± 0.1) × 10^6^ cells/mL. The cells were exposed at room temperature for 24 h after adding AiiA, His_6_-OPH, and NDM-1 enzymes.

A standard ATP reagent (Lyumtek OOO, Moscow, Russia) and the luciferin–luciferase method was used to determine the concentration of intracellular ATP by evaluating the residual concentration of viable cells in exposed samples using the published procedure [29]. The intensity of bioluminescence was recorded using a Microluminometer 3560 (New Horizons Diagnostic, Arbutus, MD, USA). The ATP concentration in the cells was measured at the beginning and end of the exposition with added enzymes.

All results are presented as the means of at least three independent experiments ± standard deviation (±SD). Statistical analysis was realized using SigmaPlot (ver. 12.5, Systat Software Inc., San Jose, CA, USA).

## 3. Results

### 3.1. Computational Modeling of Interactions of Enzymes Exhibiting Lactonase Activity with Fungal QS Molecules

Using the computational method of molecular docking, models of the interaction of enzymes selected for analysis (Table 1) with various lactone-containing signaling molecules of fungal QS were obtained. In addition to the 10 enzymes listed in Table 1, the New Delhi MBL-1 enzyme (NDM-1) (#11), representing a relatively new MBL, was also involved in studies for the presence of lactonase activity [33]. This was carried out due to the fact that a number of MBLs, including those listed in Table 1 [21,22], are known to exhibit effective lactonase activity in addition to β-lactamase activity.

In the obtained “enzyme–QS molecule” models, the characteristics of interactions were analyzed (Figure 2 and Appendix A, Table 2 and Appendix A). In addition, theoretical values of the Michaelis constant (*K*_m_) for these substrates when hydrolyzed by His_6_-OPH were calculated. For this, the equation of the dependence of the logarithms of the *K*_m_ on the affinity of substrate binding during the molecular docking of His_6_-OPH with “usual” substrates was used [34] (Appendix A).

An analysis of the values of the areas occupied by the signaling molecules of fungal QS in the active centers of the investigated enzymes was conducted using the molecular docking method. A high probability of hydrolysis (>50%) of lactone-containing signaling molecules of QS was predicted by most of the 11 hydrolytic enzymes studied. For example, multicolic acid was estimated as a possible substrate for 8 out of 11 enzymes. Multicolanic acid, γ-heptalactone, and multicolosic acid were assessed as potential substrates for seven enzymes, whereas γ-butyrolactone and butyrolactone I molecules were evaluated as possible substrates for six enzymes. At the same time, the probability of catalysis to one degree or another of all the studied QS molecules was established for all enzymes. The only exceptions were AidC, AiiB, ZEN, SsoPox, and NDM-1, for which the probability of catalysis for ≥1 of the quorum molecules was absent.

It is interesting to note that in the case of MBLs (MiM-1 and MiM-2), the probability of catalysis of all signaling molecules was high enough (>50%), with the exception of the molecule of γ-butyrolactone for MiM-2. The range of values that characterized the area occupied by lactone-containing molecules in the active centers of the enzymes with lactonase activity varied quite widely (from 0 to 98%) and depended on the enzyme. The highest occupation of the active site of lactamase MiM-1 (77–95%) by lactone-containing fungal QS molecules was noted, whereas the lowest level was revealed for lactonase AiiB (0–19%).

As a result of comparing the affinity values of the interaction of QS signaling molecules with the surface of enzymes possessing lactonase activity, it was found that the strongest binding to the surface of all enzymes was observed for the butyrolactone I molecule, and the weakest one was estimated for the γ-butyrolactone molecule.

Quorum molecules such as γ-heptalactone, butyrolactone I, and multicolic acid were noted as the most “positive” substrates for the efficient processes of the hydrolytic reaction catalyzed by the studied enzymes (Figure 3, Figure 4 and Figure 5).

### 3.2. Investigation of Catalytic Characteristics of Enzymes Exhibiting Lactonase Activity in Relation to Fungal QS Molecules

In order to experimentally confirm the results predicted on the basis of computer analysis, the hydrolysis of lactone-containing signaling molecules of γ-heptalactone and butyrolactone I by the selected enzymes (lactonase AiiA from *Bacillus thuringiensis*, lactamase NDM-1, and organophosphate hydrolase His_6_-OPH) was investigated. The catalytic characteristics (*K*_m_, *V*_max_, and action efficiency constant (*k_eff_*)) were determined (Table 3, Appendix A).

As a result of the conducted experimental studies, the possibility of effective hydrolysis of the signaling molecules of the fungal QS by the investigated enzymes was demonstrated for the first time. The hydrolysis efficiency of γ-heptalactone (*k_eff_*) decreased in the range of NDM-1 > AiiA > His_6_-OPH.

It is noteworthy that the MBL NDM-1 hydrolyzed the γ-heptalactone molecule more effectively than lactonase AiiA and His_6_-OPH (7.5 and 57.5 times, respectively). This is in addition to the fact that the possibility of hydrolysis of the fungal QS molecule by this enzyme was shown for the first time.

It was found that the combination of enzymes with each other in some cases makes it possible to significantly increase the degree of hydrolysis of γ-heptalactone. For example, in the case of the combination NDM-1/AiiA (Figure 6), the degree of hydrolysis increased up to 2.5 times compared with the individual enzymes AiiA and NDM-1.

In the case of the butyrolactone I molecule, effective hydrolysis was noted only in the case of the His_6_-OPH. At the same time, the efficiency of hydrolysis of butyrolactone I by this enzyme was eight times higher than the efficiency of the hydrolysis of γ-heptalactone (Table 3).

It should be noted that the theoretical values of *K*_m_ for γ-heptalactone and butyrolactone I (calculated from the results of molecular docking when His_6_-OPH was used for the simulation of their hydrolysis) and the experimental values were approximately equal. This once again confirms the high reliability and effectiveness of the results predicted by computational methods.

Thus, an experimental study of the catalytic characteristics of the enzymes, possessing different “natural” activity, in the hydrolytic reactions with γ-heptalactone and butyrolactone I made it possible, for the first time, to demonstrate the possible efficient hydrolysis of signaling QS molecules of fungi, as well as to reveal the high hydrolytic activity of MBLs such as NDM-1 against γ-heptalactone.

### 3.3. Investigation of Antifungal Activity of Enzymes Exhibiting Lactonase Activity

The antifungal activity of enzymes exhibiting lactonase activity (AiiA, NDM-1, and His_6_-OPH) was studied against the filamentous fungi *Rhizopus oryzae* F814, *Aspergillus niger* F679, *Trichoderma atroviride* F207, and *Fusarium solani* F819, as well as the yeast cells *Saccharomyces cerevisiae* Y-1234 and *Candida tropicalis* Y-2245 (Figure 7).

It was found that the AiiA enzyme (Figure 7A) exhibits significant antimicrobial efficacy against the fungi *F. solani* (residual ATP concentration was close to 0%), *T. atroviride* (residual ATP concentration was less than 2%), and *R. oryzae* (residual ATP concentration was less than 21%). However, in the case of *A. niger* and the yeasts *S. cerevisiae* and *C. tropicalis*, no decrease in ATP concentration was observed.

The effectiveness of the antimicrobial action of NDM-1 (Figure 7C) against the analyzed samples of microorganisms was the same as that of the AiiA enzyme. That is, there was a significant decrease in the intracellular ATP concentration in *F. solani*, *T. atroviride,* and *R. oryzae* fungal cells (residual ATP concentration was close to 0%, less than 2%, and <21%, respectively). The absence of antifungal activity was revealed in the cases of other cells.

Unlike the AiiA and NDM-1 enzymes, the His_6_-OPH (Figure 7B) had a less significant effect on changing the level of intracellular ATP in the tested cells of microorganisms. However, despite this, only in the presence of His_6_-OPH was a decrease in ATP concentration in *A. niger* cells noted (by 25%). Also, a decrease in intracellular ATP concentration in the presence of His_6_-OPH was noted for samples with *R. oryzae* (by 60%).

## 4. Discussion

It is known that different lactone-containing signaling molecules are involved in the development of the QS mechanism of resistance in the cells of a number of fungi, as well as in the case of Gram-negative bacterial cells [10,11]. However, despite this, the ways of suppressing fungal QS are poorly understood. Since one of the widespread methods of suppressing bacterial QS is the use of enzymes exhibiting lactonase activity against quorum signaling molecules, the use of molecular docking method allowed us to investigate in silico the characteristics of the interactions between various enzymes with lactonase activity and lactone-containing signaling molecules of fungal QS.

As a result, it was shown that most enzymes active against bacterial lactone-containing QS molecules are also able to effectively interact with several signaling molecules of the fungal QS. At the same time, for the first time, the possibility of interaction of various MBLs with fungal QS molecules was estimated using computer simulation. It was found that among the signaling molecules of the fungal QS studied in this work, the most “positive” substrates for enzymes, from the point of view of the hydrolytic reaction, appeared to be the molecules of γ-heptalactone, butyrolactone I, and multicolic acid.

This experimental study on the effectiveness of the catalytic action of enzymes possessing different natural substrate specificity (bacterial lactonase AiiA, organophosphate hydrolase His_6_-OPH, and lactamase NDM-1) in catalytic reactions with quorum molecules of fungi allowed us, for the first time, to demonstrate the activity of all three enzymes against the γ-heptalactone molecule and the activity of His_6_-OPH against butyrolactone I. The higher efficiency (*k_eff_*) of γ-heptalactone hydrolysis by the NDM-1 enzyme in comparison with AiiA and His_6_-OPH has been shown. Moreover, the same constant revealed for this MBL was 1.8 and 2.8 times higher than the *k_eff_* of the γ-butyrolactone molecule hydrolysis, which was established for AaL and ZEN, known as enzymes with good lactonase activity [14,15].

It should be noted that, despite the high probability of hydrolytic catalysis of butyrolactone I by NDM-1, predicted for this enzyme on the basis of molecular docking, it was not experimentally achieved. This means that, despite the fact that molecular modeling makes it possible to predict, with high reliability, the probability of a particular catalytic reaction, the theoretical and experimental effectiveness of the reaction may not always coincide. At the same time, however, it is also worth noting that the Michaelis constant values in the hydrolysis reactions of the fungal quorum molecules by the enzyme His_6_-OPH, obtained experimentally, coincided with the values calculated theoretically for two substrates (γ-heptalactone and butyrolactone I) (Table 3).

This study on the antifungal activity of enzymes in relation to various fungi and yeasts allowed us to confirm the results of studies obtained using computer methods and studies on the catalytic properties of the enzymes. The high antifungal efficacy of action of AiiA and NDM-1 against *F. solani*, *T. atroviride,* and *R. oryzae* cells has been shown. It is noteworthy that both enzymes, AiiA and NDM-1, proved to be equally effective against the cells of the tested fungal cultures. This is despite the fact that, unlike the AiiA enzyme, which mainly hydrolyzes lactone-containing compounds, the main substrates of the NDM-1 enzyme are compounds containing a β-lactam ring. The antifungal activity of the His_6_-OPH enzyme against *A. niger* and *R. oryzae* cells was also established.

In almost all fungi, the main cell wall has a branched glucan, which is associated with chitin. In general, the composition of the cell wall in different filamentous fungi is approximately similar. There is a variation in the ratio of the main components, which depends on the age of the mycelium, its morphology, and the substrate used for the accumulation of fungal biomass [35,36]. It can be assumed that this particular variation somehow leads to different effects of the same enzymes on the different fungal cells, as well as to differences between the actions of the enzymes themselves.

Interestingly, almost all of the studied enzymes turned out to be ineffective as antifungals against yeast cells, even considering the fact that a slight decrease in the residual concentration of ATP was noted (by 3–4%) in the case of His_6_-OPH. This may be due to the obvious differences in the structure of the cell wall of various fungi and yeasts, as well as to the existing differences in QS activity between the yeasts themselves. For example, it is known that yeasts of the *Candida* and *Saccharomyces* species have an outer cell wall containing highly mannosylated glycoproteins that cover the inner wall. At the same time, α-(1,3)-glucan, which plays an important role in the organization of the cell wall, is absent in the cells of these yeast species [37,38].

Presumably, the notably lower concentrations of lactone-containing molecules in the yeast cells compared to fungi may be another reason for the lack of effectiveness of the enzymes in relation to yeast cells. It is known that fungi are capable of producing various types of lactones, and therefore, they are often used as producers of such compounds in biotechnological processes [39,40]. However, the biosynthesis of QS lactones is a complex process in fungi that has not yet been completely studied. For example, some microorganisms are able to independently produce lactones without introducing any precursors into the medium, whereas for others, the introduction of fatty acids or their derivatives into the medium for cell cultivation is a necessary condition for lactone synthesis [40].

Thus, it is possible that a combination of changes in the composition of nutrient media for fungi and the introduction of enzymes that catalyze the destruction of lactone-containing signaling molecules of fungal QS into these media may give improved results in further studies on suppressing the growth of fungal cultures in comparison with the use of only lactone-hydrolyzing enzymes, as was carried out in this work. Along with this, the results obtained in this work can serve as a basis for the development of new enzyme-based antifungals without the application of antibiotics. Antifungals based on these enzymes can be used as dressings, gels, or sprays for the treatment of various skin injuries in humans or animals associated with both bacterial and fungal infections for the development of improved wound-healing materials. In addition to medical use, preparations based on these enzymes can also be used in the food industry as additives to packaging materials to prevent the development of resistant populations of microorganisms and the production of toxic metabolites. In perspective, this approach looks similar to known lysozymes used against bacterial cells as an enzymatic antimicrobial analogue of antibiotic blockbusters [41].

## 5. Conclusions

Thus, in this work, for the first time, the presence of catalytic potential in enzymes with different natural activity (lactonase AiiA, organophosphate hydrolase His_6_-OPH, and MBLs such as NDM-1) in relation to lactone-containing molecules of fungal QS was established. The possibility of efficient use of these enzymes to suppress the QS of fungi was shown using samples of different fungal and yeast cultures. The presence of high antifungal activity in AiiA and NDM-1 enzymes against cells of various fungi has been established. It was also shown, for the first time, that the combination of these enzymes with each other provides a significant improvement in both their catalytic and antifungal properties. The results obtained in this study suggest that the investigated enzymes, AiiA, His_6_-OPH, and NDM-1, can be successfully used to suppress fungal QS in various microorganisms, both individually and in combination with each other. Further, these enzymes can be investigated as components of antifungal compositions with various antifungals to increase the effectiveness of the latter.

## Figures and Tables

**Figure 1 biomolecules-14-00383-f001:**
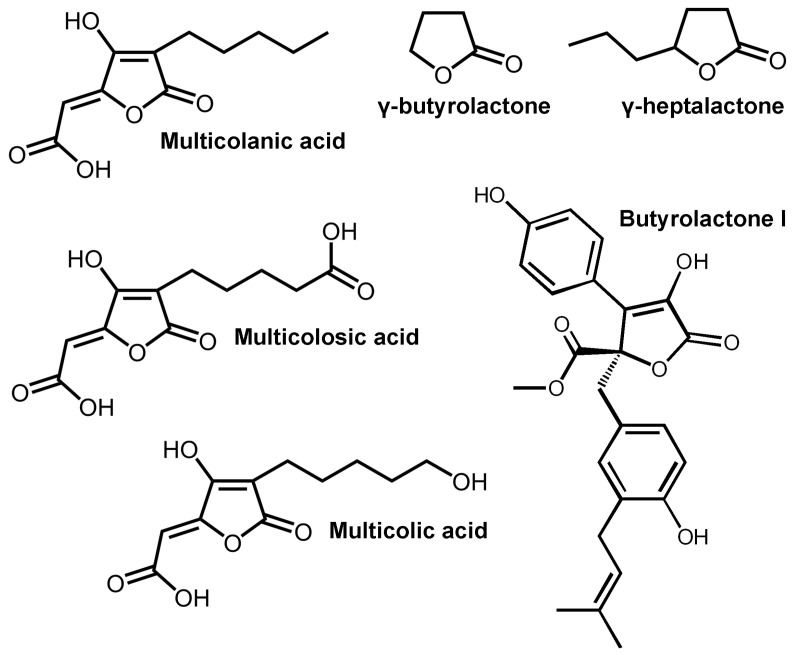
Lactone-containing molecules of fungal QS.

**Figure 2 biomolecules-14-00383-f002:**
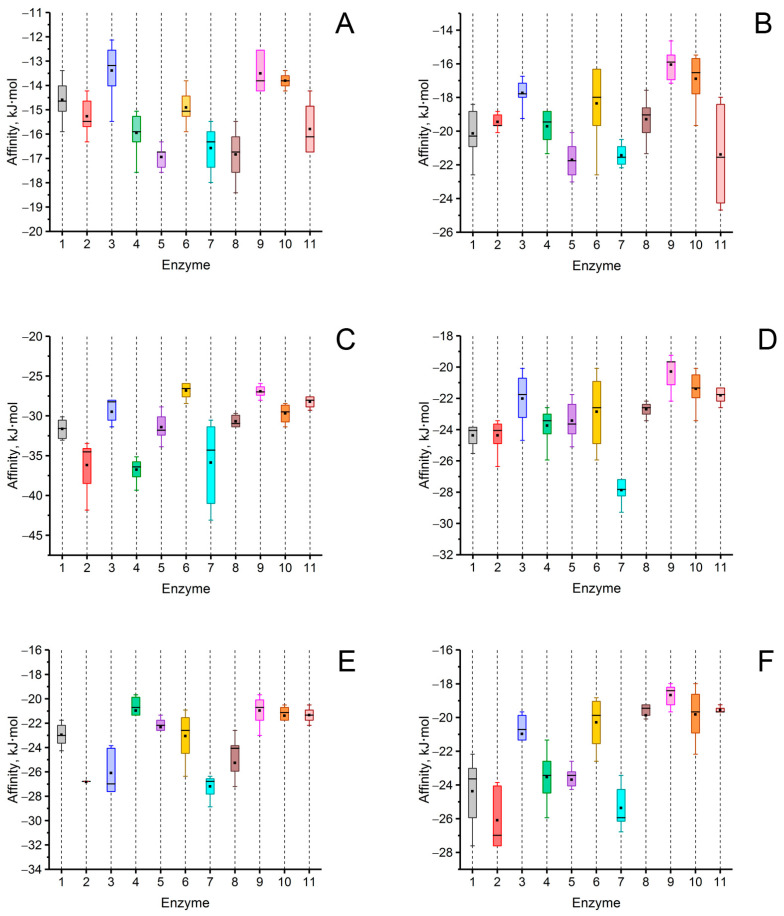
Affinity of signaling molecules of fungal QS such as γ-butyrolactone (**A**), γ-heptalactone (**B**), butyrolactone I (**C**), multicolanic acid (**D**), multicolic acid (**E**), and multicolosic acid (**F**) to the surface of 11 selected enzymes exhibiting lactonase activity. The rectangles correspond to interquartile range (between 75th and 25th percentile). Horizontal lines (─) indicates median value. ■ symbol indicates mean value. The numbers of enzymes correspond those mentioned in Table 1 (number 11 corresponds to the enzyme NDM-1). The colors correspond to those applied in Table 1 to indicate the enzymes used.

**Figure 3 biomolecules-14-00383-f003:**
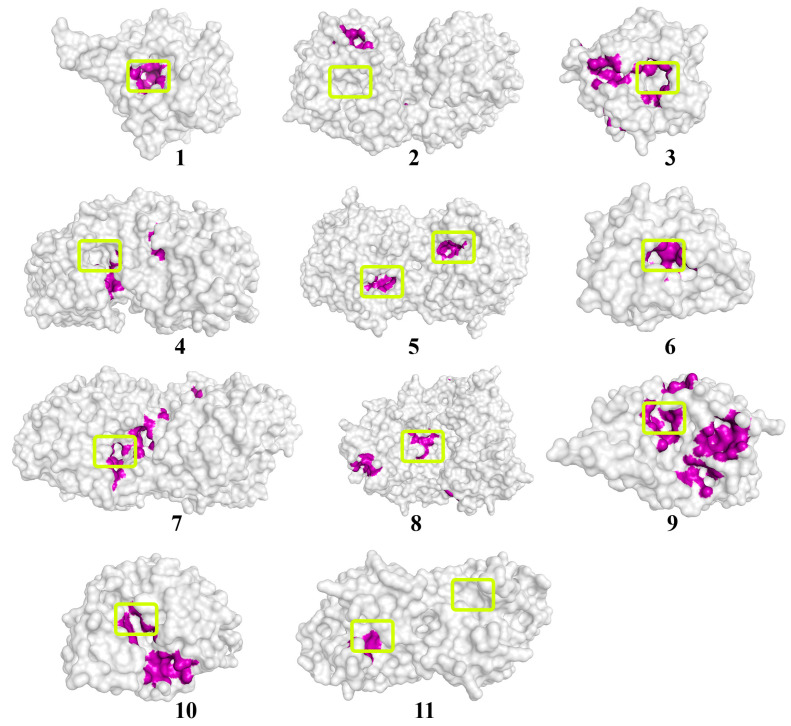
The location of the γ-heptalactone molecule on the surface of enzymes exhibiting lactonase activity. The molecular surface of the enzymes is in a translucent gray color. The atoms located within 4 Å of any γ-heptalactone atom and the corresponding molecular surface of enzymes are colored purple. The entrances to the active sites of the enzymes are highlighted with light-green boxes. The numbers of enzymes from 1 to 10 correspond those mentioned in Table 1, and enzyme # 11 corresponds to NDM-1.

**Figure 4 biomolecules-14-00383-f004:**
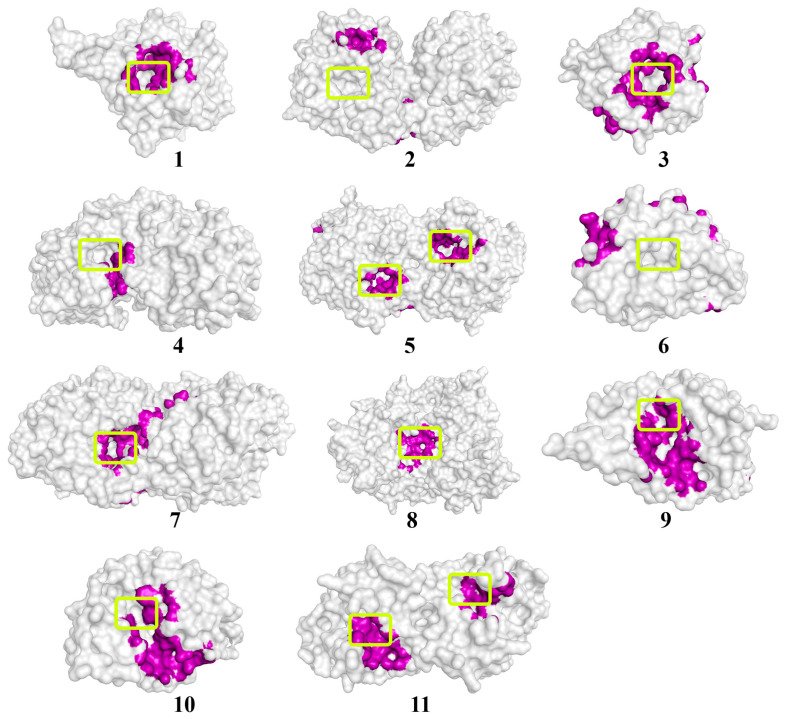
The location of the butyrolactone I molecule on the surface of enzymes exhibiting lactonase activity. The molecular surface of the enzymes is in a translucent gray color. The atoms located within 4 Å of any butyrolactone I atom and the corresponding molecular surface of enzymes are colored purple. The entrances to the active sites of the enzymes are highlighted with light-green boxes. The numbers of enzymes from 1 to 10 correspond those mentioned in Table 1, and enzyme # 11 corresponds to NDM-1.

**Figure 5 biomolecules-14-00383-f005:**
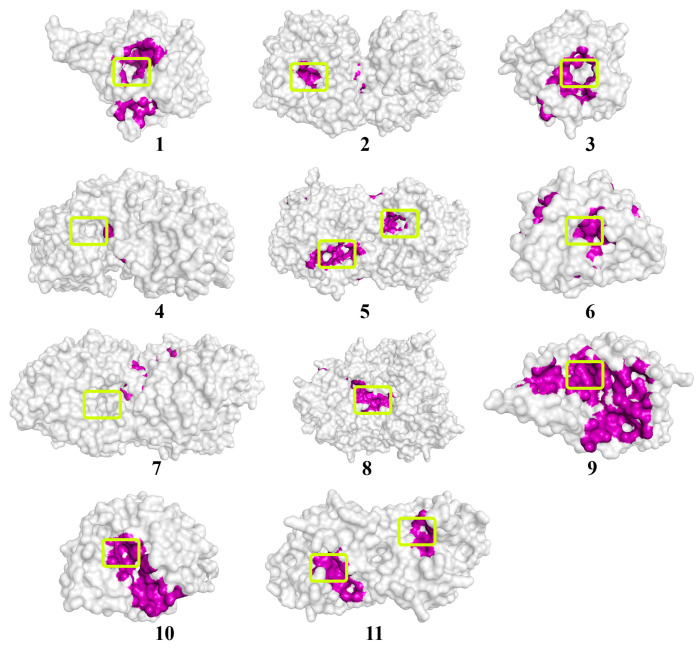
The location of the multicolic acid molecule on the surface of enzymes exhibiting lactonase activity. The molecular surface of the enzymes is in a translucent gray color. The atoms located within 4 Å of any multicolic acid atom and the corresponding molecular surface of enzymes are colored purple. The entrances to the active sites of the enzymes are highlighted with light-green boxes. The numbers of enzymes from 1 to 10 correspond those mentioned in Table 1, and enzyme # 11 corresponds to NDM-1.

**Figure 6 biomolecules-14-00383-f006:**
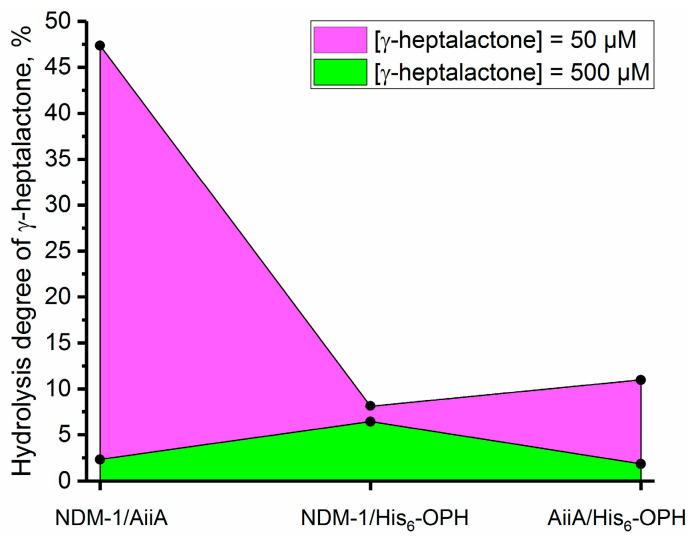
The degree of hydrolysis of the γ-heptalactone molecule under the action of enzyme combinations of NDM-1/AiiA, NDM-1/His_6_-OPH, and AiiA/His_6_-OPH at 37 °C for 3 h. The concentration of γ-heptalactone in the control medium in the absence of the enzyme was assumed to be 100%.

**Figure 7 biomolecules-14-00383-f007:**
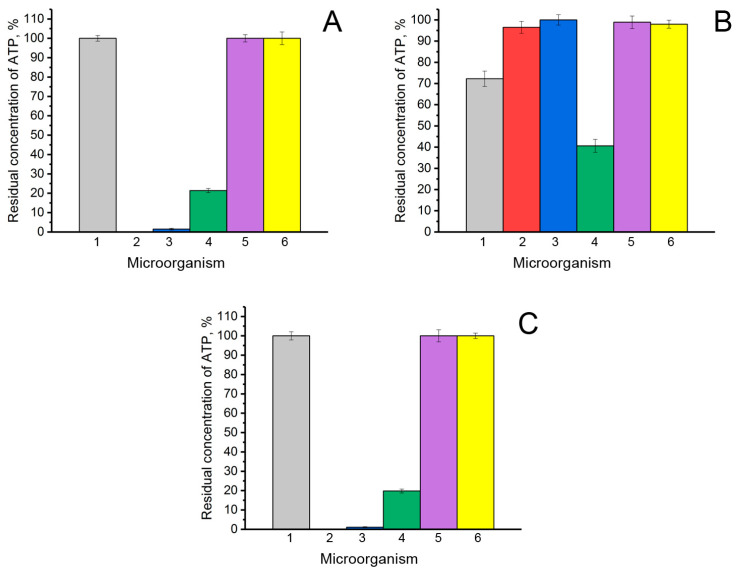
Residual concentration of intracellular ATP as an indicator of the antimicrobial efficacy of the antifungal action of the enzymes AiiA (**A**), His_6_-OPH (**B**), and NDM-1 (**C**) on the cells of filamentous fungi and yeast cells: 1—*A. niger* (
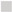
), 2—*F. solani* (
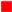
), 3—*T. atroviride* (
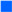
), 4—*R. oryzae* (
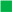
), 5—*S.cerevisiae* (
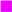
), and 6—*C. tropicalis* (
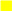
). The concentration of ATP at the initial time of experiment in control samples in the corresponding cells without the addition of enzymes was taken as 100%.

**Table 1 biomolecules-14-00383-t001:** Different lactonases and MBLs hydrolyzing lactone-containing QS molecules.

#	Enzyme, Origin	PDB ID	Substrates	[References]
1	His_6_-OPH from *Pseudomonas diminuta*	1qw7 (OPH)	C4-C8-HSL, 3OC8-C12-HSL	[12,16]
2	AaL from *Alicyclobacillus acidoterrestris*	6cgz	C6-C10-HSL, 3OC12-HSL, γ-butyrolactone, δ-lactones	[14]
3	Zearalenon lactonase (ZEN) from *Clonostachys rosea*	3wzl	C4-C10-HSL, γ-lactones, δ-lactones	[15]
4	AidC from *Chryseobacterium* sp.	4zo2	C6-C12-HSL, 3OC6-C12-HSL	[17]
5	AiiA from *Bacillus thuringiensis*	2a7m	C4-C10-HSL, 3OC4-C12-HSL, 3OHC4-HSL	[18,19]
6	AiiB from *Agrobacterium tumefaciens*	2r2d	C4-C10-HSL, 3OC6-C8-HSL	[20]
7	MIM-1 from *Novosphingobium pentaromativorans*	6auf	C4-C12-HSL, 3OHSL	[21,22]
8	SAM-1(or MIM-2) from *Simiduia agarivorans*	6mfi		[21,22]
9	SsoPox from *Sulfolobus solfataricus*	2vc7	C6-C12-HSL, 3OC6-C12-HSL	[23]
10	PvdQ from *Pseudomonas aeruginosa*	4m1j	C7-C14-HSL, 3OC10-C14-HSL, 3OHC12-C14-HSL	[24]

**Table 2 biomolecules-14-00383-t002:** Analysis and comparison of the data presented in Appendix A, based on the values of the areas occupied by the molecules of fungal QS (1—γ-butyrolactone, 2—γ-heptalactone, 3—butyrolactone I, 4—multicolanic acid, 5—multicolic acid, and 6—multicolosic acid) in the active centers on the surface of the investigated enzymes.

Enzymes	1	2	3	4	5	6
Aal						
AidC						
AiiA						
AiiB						
His_6_-OPH						
ZEN						
SsoPox						
PvdQ						
MIM-1						
MIM-2						
NDM-1						

Note: The red color indicates the absence of the probability of catalysis (the molecule is outside the active center). Yellow color means that probability of catalysis is less than 50% (the active center is partially occupied). Green means a high probability of catalysis (>50%).

**Table 3 biomolecules-14-00383-t003:** Catalytic characteristics of enzymes with lactonase activity in hydrolytic reactions of the fungal QS molecules such as γ-heptalactone and butyrolactone I.

Enzyme	*K*_m_, µM	*V*_max_/E_0_, s^−1^	*k_eff_*, 10^3^·s−^1^·M^−1^
**γ-heptalactone**			
AiiA	90.4 ± 10.2	4.9 ± 0.9	54 ± 16
His_6_-OPH	316 (theoretical)/362 ± 22 (experimental)	4.4 ± 0.7	12 ± 3
NDM-1	43 ± 3	17 ± 3	403 ± 101
**butyrolactone I**			
His_6_-OPH	30 (theoretical)/29 ± 4 (experimental)	1.7 ± 0.09	58 ± 11

## Data Availability

Data are contained within the article.

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
