# Peer review of "Enzymes with Lactonase Activity against Fungal Quorum Molecules as Effective Antifungals"

_biomolecules, 2024, doi:10.3390/biom14030383_

Round 1

Reviewer 1 Report

Comments and Suggestions for Authors

Please find attached report

Author Response

Dear Reviewer,

We are grateful to you for the suggestions allowing us the improving of our manuscript. Please, see our comments to your remarks and the revised text of the paper:

Responses to the Comments of Reviewer:

The article by Efremenko et. al. describes action of lactonase enzymes exhibiting lactonase activity against fungal signaling molecules as similar to of Quorum quenching in bacteria. 11 different enzymes were studied against 6 different lactone containing molecules for QS. They demonstrated higher antifungal activity of enzymes AiiA and NDM-1 among the 6 different cells of various fungi and yeast has been established. Authors claimed that the the results obtained in this study the investigated enzymes AiiA, His6-OPH and NDM-1 can be successfully used to suppress the fungal QS in various microorganisms, both individually and in combination with each other.

Manuscript is well written. Data supports the conclusions. Therefore, I recommend this article to publish after minor revision.

Response: We are sincerely grateful to you for your attentiveness to the text of the article! We took all comments and suggestions into account and made appropriate changes to the text.

Table1: Add column to number the enzymes (1 to 11). Adding color codes is also recommended for the better understanding of next analytical data (in fig-2).

Response: We have added additional column to Table 1 in order to number the enzymes and color code corresponding to colors in Figure 2.

Figure -2: Figure legends at the bottom E is repeated. It needs to change to F.

Response: It has been corrected.

Table 2: since it shows only the range by the color code, table can be re-drawn with smaller column widths.

Response: We modified the Table 2.

Fig- 3, 4, 5: molecular docking figure legend should also describe thee highlighted green square indicators.

Response: It has been corrected.

Fig- 6: what are the color code and concentration is for?

Response: It has been corrected.

Line -306-307: Values seems exchanged across. Not matching with the graphs in fig-7.

Response: It has been corrected.

All reference presented should be in consistent format, Example- ref 39-journal name is not Italic.

Response: It has been corrected.

In SI: GS data need to explain (Obtained Mass and the calculated mass of corresponding compound).

Response: It has been explained.

With high respect and good wishes,

Authors of the manuscript.

Reviewer 2 Report

Comments and Suggestions for Authors

Line 105: Accession number of PON2 should be included. And the percentage of identity, similarities between PON1 and PON2 shall be included in either the results or in method section. Reliability of I-Tasser model? Structure validation must be done with the Ramachandran plot. Proper structural input is essential for the molecular docking studies. Why do authors attempt for PON2 modelling, since there nowhere in the manuscripts deals with the protein PON2?

Since the base of this study is molecular docking, interaction figures can be included in the supplementary file.

In additional molecular dynamics simulations could provide the information about the molecules binding affinity and there occupancy in the cavity.

Experimental studies supports the computational predictions.

A statement about how the identified enzymes could be used for the treatment purposes would be required in the last paragraph of the discussion.  

Comments on the Quality of English Language

The written text in the materials and methods section has more similarity with the published articles, which should be rewritten.

Certain statements are quite long and complex, hence language can be checked for better readability for readers.  

Author Response

Dear Reviewer,

We are grateful to you for the suggestions allowing us the improving of our manuscript. Please, see our comments to your remarks and the revised text of the paper:

Line 105: Accession number of PON2 should be included. And the percentage of identity, similarities between PON1 and PON2 shall be included in either the results or in method section. Reliability of I-Tasser model? Structure validation must be done with the Ramachandran plot. Proper structural input is essential for the molecular docking studies. Why do authors attempt for PON2 modelling, since there nowhere in the manuscripts deals with the protein PON2?

Response: We sincerely thank you for your careful reading of the manuscript and your comments and suggestions! We have tried to take into account all of them and have added additional details and some corrections into the text.  Since the information about PON2 protein is irrelevant to this study, we have removed it from the text. We do apologize for causing misunderstanding.

Since the base of this study is molecular docking, interaction figures can be included in the supplementary file.

Response: We have added additional figure (Figure S1) with 3D structures of lactone-containing fungal quorum molecules, used for the interaction with enzymes, to the supplementary file. Figures showing enzyme-ligand interactions are already present in file with Supplementary materials (Figures S2-S12).

In additional molecular dynamics simulations could provide the information about the molecules binding affinity and their occupancy in the cavity.

Response: Thank you very much for your suggestion! We understand that both molecular docking and molecular dynamic simulations are very important techniques to understand the interactions of ligand molecules with protein. In general molecular docking simulations allow you to predict the interactions between protein and ligand molecules “here and now” (in the moment), while molecular dynamic simulations predict the interaction between them “in time” (the real mode of interactions), allowing to check the stability of the protein-ligand complexes by analyzing a number of physical-chemical parameters. In this regard, the choice of one method or another (or both) primarily depends on the goals of the study. In this study, our main goal was to determine the presence or the absence of the possibility of catalysis of lactone-containing molecules of fungal QS by selected “lactonases”. In this regard, the use of the molecular docking method was sufficient for us in order to predict the interaction of quorum molecules with enzymes and select potential catalysts for these molecules.

Experimental studies support the computational predictions.

Response: Thank you for your comment!

A statement about how the identified enzymes could be used for the treatment purposes would be required in the last paragraph of the discussion. 

Response: We have added this additional text to the last paragraph of the discussion: “Antifungals based on these enzymes can be used as dressings or sprays for the treatment of various skin injuries in humans or animals associated with both bacterial and fungal infections, for the development of improved wound-healing materials. In addition to medical use, preparations based on these enzymes can also be used in the food industry as additives to packaging materials to prevent the development of resistant populations of microorganisms and the production of toxic metabolites.”

Comments on the Quality of English Language: The written text in the materials and methods section has more similarity with the published articles, which should be rewritten.

Response: It has been rewritten.

Certain statements are quite long and complex, hence language can be checked for better readability for readers.

Response: We have checked and edited the English language throughout the article to simplify all complex sentences and make it clearer as much as possible.

With high respect and good wishes,

Authors of the manuscript.
